# Evolution of *kdr* haplotypes in worldwide populations of *Aedes aegypti*: Independent origins of the F1534C *kdr* mutation

Luciano Veiga Cosme[1], Andrea Gloria-Soria[1,2], Adalgisa Caccone[1], Jeffrey Robert Powell[1], Ademir Jesus Martins[3,4] *

**1** Yale University, New Haven, CT, United States of America, **2** Center for Vector Biology & Zoonotic Diseases. The Connecticut Agricultural Experiment Station, New Haven, CT, United States of America, **3** Laboratório de Fisiologia e Controle de Artrópodes Vetores, Instituto Oswaldo Cruz/ FIOCRUZ, Av Brasil, Rio de Janeiro, RJ, Brazil, **4** Instituto Nacional de Ciência e Tecnologia em Entomologia Molecular, INCT-EM, UFRJ, Rio de Janeiro, RJ, Brazil

* ademirjr@ioc.fiocruz.br

**Data Availability Statement:** All data generated or analyzed during this study are included within the manuscript and its Supporting Information files, as well as in the Sequence Read Archive (SRA

## Abstract

*Aedes aegypti* is the primary vector of dengue, chikungunya, Zika, and urban yellow fever. Insecticides are often the most effective tools to rapidly decrease the density of vector populations, especially during arbovirus disease outbreaks. However, the intense use of insecticides, particularly pyrethroids, has selected for resistant mosquito populations worldwide. Mutations in the voltage gated sodium channel ($Na_V$) are among the principal mechanisms of resistance to pyrethroids and DDT, also known as "knockdown resistance," *kdr*. Here we report studies on the origin and dispersion of *kdr* haplotypes in samples of *Ae. aegypti* from its worldwide distribution. We amplified the IIS6 and IIIS6 $Na_V$ segments from pools of *Ae. aegypti* populations from 15 countries, in South and North America, Africa, Asia, Pacific, and Australia. The amplicons were barcoded and sequenced using *NGS* Ion Torrent. Output data were filtered and analyzed using the bioinformatic pipeline Seekdeep to determine frequencies of the IIS6 and IIIS6 haplotypes per population. Phylogenetic relationships among the haplotypes were used to infer whether the *kdr* mutations have a single or multiple origin. We found 26 and 18 haplotypes, respectively for the IIS6 and IIIS6 segments, among which were the known *kdr* mutations 989P, 1011M, 1016I and 1016G (IIS6), 1520I, and 1534C (IIIS6). The highest diversity of haplotypes was found in African samples. *Kdr* mutations 1011M and 1016I were found only in American and African populations, 989P + 1016G and 1520I + 1534C in Asia, while 1534C was present in samples from all continents, except Australia. Based primarily on the intron sequence, IIS6 haplotypes were subdivided into two well-defined clades (A and B). Subsequent phasing of the IIS6 + IIIS6 haplotypes indicates two distinct origins for the 1534C *kdr* mutation. These results provide evidence of *kdr* mutations arising *de novo* at specific locations within the *Ae. aegypti* geographic distribution. In addition, our results suggest that the 1534C *kdr* mutation had at least two independent origins. We can thus conclude that insecticide selection pressure with DDT and more recently with pyrethroids is selecting for independent convergent mutations in $Na_V$.

Accession number: PRJNA579141, Biosample number: SAMN13092781). The sequences of the haplotypes were submitted to Genebank (accession numbers: MN602753-MN602796).

**Funding:** This study was funded by the National Institute of Health (NIH, Grant no. UO1 AI115595) to JP, the Instituto Nacional de Ciência e Tecnologia em Entomologia Molecular (INCT-EM, Grant n0 465678/2014-9) to AJM, and the Fundação de Amparo à Pesquisa do Estado do Rio de Janeiro (FAPERJ, Grants no E-26/203.177/2016, E-26/201.836/2017) also to AJM. The funders had no role in study design, data collection and analysis, decision to publish, or preparation of the manuscript.

**Competing interests:** The authors have declared that no competing interests exist.

## Author summary

Insecticide resistance is a global threat for the control of *Aedes aegypti*, the mosquito vector of aboviruses such as dengue, chikungunya and Zika. Mutations in the voltage gated sodium channel ($Na_V$), known as *kdr*, are one of the principal mechanisms related to resistance to pyrethroids, the class of insecticide most employed worldwide inside and around residences. We investigate whether the same *kdr* mutations found in *Ae. aegypti* populations from distinct regions of the world have a common origin and subsequently dispersed or if they emerged in unrelated populations at distinct moments. By evaluating the sequences of two fragments of the $Na_V$ gene, obtained from DNA collections of *Ae. aegypti* from several countries, we found at least two independent origins for the F1534C *kdr* mutation in American, African and Asian populations. There was no evidence for multiple origins of the common *kdr* mutations V1016I and P989S + V1016G, which were exclusive to American and Asian populations. Our results increase our knowledge of insecticide resistance evolution in one of the main arboviral mosquito vectors of major global diseases.

## Introduction

*Aedes aegypti* is considered one of the most successful invasive species worldwide. Originally from Africa and now found throughout the tropics and subtropics, i t has been divided into two subspecies *Aedes aegypti aegypti* (Aaa) and *Aedes aegypti formosus* (Aaf). The subspecies Aaa, hereafter referred simply as *Ae. aegypti*, spread to the five continents, starting from Africa to the Americas during the human slave trade that began in the 1500's, and later (1800's) from the Americas to Asia and Australia [1]. *Ae. aegypti* is generally referred to as the yellow fever mosquito, as it was the vector of yellow fever virus in the urban cycle of the disease in the Americas until the first half of the last century and more recently in West Africa [2]. Vaccination against yellow fever and intense vector control eliminated the urban cycle of the disease in most countries by mid-20[th] Century [3]. However, decreasing entomological surveillance along with rapid increases in urbanization and demographic changes has led to the re-establishment of this species. Currently *Ae. aegypti* occurs on six continents [4], reaching high densities in urban centers of tropical and subtropical cities and acts as the primary vector of viruses causing dengue, chikungunya and, Zika, as well as re-emerging yellow fever [5]. Today, the risk of epidemic arboviral diseases is the highest in history [6] and it is predicted that in response to increased urbanization and climate change, 2.25 billion more people (or 60% of global population) will be at risk of dengue in 2080 compared to 2015 [7].

The first line of effective control of *Ae. aegypti* should be improvement of sanitary infrastructure, with insecticides employed as a complementary tool during high density infestation and arboviruses epidemic seasons [8]. However, insecticides have been extensively used as the primary front of action, even during low-density seasons, increasing selection for resistance. Insecticide resistance is now recognized as a major threat for the control of arboviral diseases and has likely contributed to their re-emergence and spread. Important knowledge gaps remain on mosquito insecticide resistance, including its global distribution, dynamics, mechanisms, fitness costs, and impact on vector control [9].

In contrast to the numerous insecticidal compounds available for agriculture, only a few classes of insecticides are permitted for public health purposes, such as larvicides in drinkable water tanks. Fewer are approved for spatial spraying against adult mosquitoes, for example,

only pyrethroids are recommended for indoor spraying [10] and treatment of mosquito nets [11]. Pyrethroids are the most frequently used insecticides globally, due to their low toxicity to mammals, limited impact on the environment, and rapid effects on insects. This rapid action is known as the *knockdown effect*, which includes intense nerve firings, followed by paralysis and death. DDT has similar effects, as both compounds interact with the voltage gated sodium channel ($Na_V$), a transmembrane protein of neuron axons [12]. Common mechanisms of resistance against pyrethroids include an increased expression of detoxifying enzyme genes, especially P450 *cyp* genes, or single nucleotide polymorphisms (SNPs) in the pyrethroid target $Na_V$ gene, known as *kdr* (*knockdown resistance*).

The *Ae. aegypti* $Na_V$ protein contains 2061 aa (amino acids) [13], with four similar domains (I-IV), each composed of six transmembrane segments (S1-S6) Fig 1. The first *kdr* mutation was identified in *Musca domestica*, at the 1014 aa site in the IIS6 segment [14]. In several other insects, including mosquitoes of the genus *Anopheles* and *Culex*, *kdr* mutations were identified at the same site. Mutations at several other $Na_V$ sites have also been classified as *kdr* mutations due to a correlated resistance to pyrethroids and/or DDT. The principal *kdr* mutations described in *Ae. aegypti* are V410L (IS6), P989S (in the link IIS5-S6), I1011M, V1016I and V1016G (IIS6), T1520I and F1534C (IIIS6) [9].

Some of these mutations, such as F1534C, are present in multiple populations around the world. This could be the due to a single origin event that spread widely or a consequence of multiple independent events. The same mutation could arise independently if there were functional restrictions on the number of resistance-inducing substitutions that would retain proper physiological function of the sodium channel, as is the case of the L1014F in *An. gambiae* [15]. Here, we use a next generation sequencing approach to explore the variation in the nucleotide regions encoding part of the IIS6 and IIIS6 segments of the $Na_V$ of *Ae. aegypti*, where the most important *kdr* mutations are found. We then reconstruct the evolutionary relationships among haplotypes to determine the origin of worldwide *kdr* mutations in *Ae. aegypti*.

## Methods

The Ion Torrent PGM platform (Thermo Fisher Scientific Inc.) was used to sequence ~400 bases of the IIS6 and IIIS6 segments of the voltage gated sodium channel gene of *Aedes aegypti* (*AaNa_V*), which contain most of the described *kdr* mutations.

### DNA amplification

*Ae. aegypti* DNA was obtained from the nucleic acid collection maintained at the Powell lab at the Ecology and Evolutionary Biology Department of Yale University; generated from previous studies [16–18]. An aliquot of 1 μL (~30 ng) from single mosquito DNA preparations from the same population samples were pooled. Each pool contained an average of 30 samples (see S1 Table). Pooled DNA was then used to amplify segments IIS6 and IIIS6 of the $Na_V$ gene in separate PCR reactions, with the Phusion High-Fidelity DNA Polymerase PCR kit (New England BioLabs). Each reaction consisted of 40 μL, containing 1X Phusion Hf buffer, 200 μM dNTP, 3% DMSO, 0.4 U Polymerase, 1.0 μM of each primer, 2 μL of pooled DNA and $H_2O$ *q.s.* 40 μL. The primers employed were *5para6*: CGGGTATTATGCGGCGAGTG x *3para3*: TGGA-CAAAAGCAAGGCTAAG for the IIS6 segment, and *AegNaVfor8*: GTGGGAAAGCAGCC-GATTCGC x *AegNaVrev6*: TGTTGAACCCGATGAACAAC for IIIS6 segment (all primers described in 5'to 3'orientation) (Fig 1). Thermocycling conditions were 93°C/30"followed by 35 cycles of 98°C/15", 60°C/30" and 72°C/60", with a final extension of 72°C/5', for both reactions. The PCR products were run on a 3% agarose LE electrophoresis gel, and 400 bp bands were excised and purified with the QIAquick PCR Purification Kit (QIAGEN), following

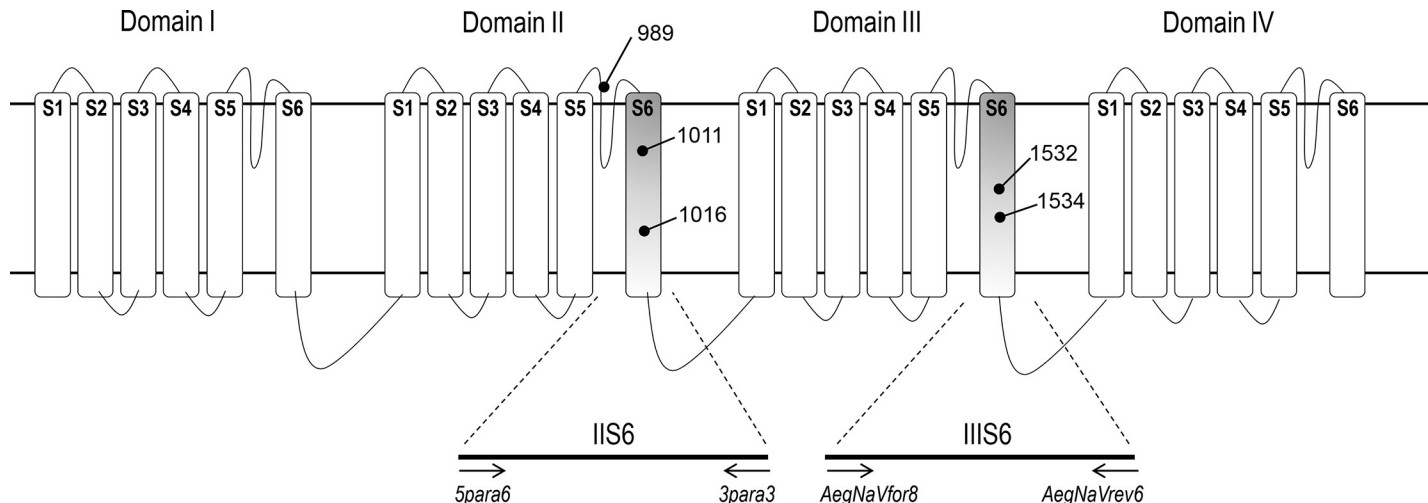

**Fig 1. Scheme of the voltage gated sodium channel.** The IIS6 and IIIS6 segments are highlighted, indicating the primers (arrows showing orientation) used to amplify their corresponding DNA regions in this study. The *kdr* mutation sites previously found in *Aedes aegypti* in these segments (989, 1011, 1016, 1532 and 1534) are also indicated.

manufacturer instructions. Purified DNA was quantified in a 2100 Bioanalyzer instrument (Agilent Technologies Inc.). Purified PCR amplicons for the two Na$_V$ segments from the same population were pooled in equimolar concentrations for library preparation.

## NGS Library preparation

We barcoded each population using the Ion Plus Fragment Library Kit (Thermo Fischer Scientific Inc.). Briefly, the pooled amplicons of each population were end repaired and purified with magnetic AMPure XP beads (Agencourt, Beckman Coulter), before ligation to adapters (Ion P1 Adapter and Ion Xpress Barcode X) and nick repair. The list of barcode sequences and their respective populations are presented in S1 Table. Libraries were then purified with magnetic beads and quantified with a High Sensitivity DNA chip in a 2100 Bioanalyzer (Agilent Technologies Inc.). Next, libraries were PCR amplified with Platinum PCR SuperMix High fidelity (100 μL), Library Amplification Primer Mix (5 μL), unamplified library (10 μL) and low TE buffer (15 μL), in five cycles, according to the conditions indicated in the manufacturer protocol. Amplified barcoded libraries were purified with magnetic beads and molar concentrations measured with the High Sensitivity DNA Kit (Agilent Technologies Inc.) in the Bioanalyzer. We then pooled 100 pM of each library to prepare the sequencing template. Two pooled templates were prepared, the first with library containing barcodes 1 to 48 and the second with 49 to 96.

## Enriched template preparation by emulsion PCR (emPCR)

First, the template-positive Ion Sphere Particle (ISP) was prepared with the kit Ion PGM Template OT2 400 kit (ThermoFischer), using the Ion OneTouche 2 Instrument. Then, the resulting template-positive ISP suspension was enriched in the Ion OneTouch ES system, continuing the procedures indicated in the same kit. Both procedures followed the manufacturer protocol.

## Sequencing

The enriched template-positive ISP was sequenced using the PGM Sequencing 400 kit (Thermo Fischer Scientific Inc.), as per manufacturer protocol. Two runs were performed, one for each pooled library template, both with an Ion 318 Chip v2.

## Data analysis

Raw sequencing data was de-multiplexed, clustered and analyzed using the software package SeekDeep [19] (http://baileylab.umassmed.edu/SeekDeep). See details in S1 Text. All data generated or analyzed during this study are included in this published article and the Sequence Read Archive available at www.ncbi.nlm.nih.gov/sra (SRA Accession number: PRJNA579141, Biosample number: SAMN13092781). Haplotypes were aligned by Muscle in the Geneious software V 11.1.5 [20]. The alignments were submitted to a TCS analysis in PopArt [21] to reconstruct the haplotype networks. Phylogenetic trees were built employing the maximum likelihood model in MEGA 7 [22].

We first evaluated genetic diversity, built the haplotypic network, and performed phylogenetic analyses for each segment independently. The haplotypes were named as 2s6 or 3s6 (in reference to IIS6 and IIIS6 segments, respectively), followed by an identifying number. As evident in the Results section, the IIS6 haplotypes were grouped into A or B clades. Accordingly, we added A or B, following 2s6, to the names of IIS6 haplotypes. Although IIS6 and IIIS6 segments are ~ 44.5 Kb apart, since linkage disequilibrium in *Ae. aegypti* is about 50–80 Kb [distance at which correlation is halved from maximum [23, 24]], we expect variation in the two segments to be correlated. As we could not phase the haplotypes when both segments were polymorphic, haplotypes phase could be determined only in populations monomorphic for one of the two segments. In these cases, phased haplotypes were noted with the prefix "*Phased*" followed by their respective IIS6 and IIIS6 haplotype designations.

## Temporal variation

As sequences of several *Ae. aegypti* populations from Brazil spanning 2001 to 2015 were available, we evaluated whether the frequencies of the observed $Na_V$ haplotypes increased or decreased over time in this country.

# Results

## Quality filtering

After quality filtering we obtained IIS6 and IIIS6 segment sequences for 79 and 71 populations, respectively. Specifics on the number of sequences used for each locality are listed in S2 Table.

## Haplotype analyses

**Spatial analyses.** Analyses of the IIS6 segment resulted in 26 haplotypes, ranging from 324 to 352 bp, among which 16 were exclusive to a single population (private alleles), all in Africa (S3 Table). The sequences of these haplotypes were submitted to Genebank (accession numbers: MN602753-MN602778) and their frequencies per locality are reported in S3 Table and S1 Fig. An alignment of the haplotypes (S2 Fig) revealed that the intron had insertions and deletions (indels). The maximum likelihood tree groups these haplotypes into two well supported clades (S3 Fig), Clade A (N = 6) and Clade B (N = 20), consistent with previous reports [25]. Eighteen of the 20 haplotypes in Clade B are exclusive to Africa. Exon regions were conserved, with four non-synonymous substitutions in codons 989 (TCC/CCC: S/P), 1011 (ATG/ATA: I/M) and 1016 (GTA/ATA/GGA: V/I/G). All non-synonymous substitutions have been

previously described as *kdr* mutations [9]. There were four codons with synonymous substitutions: amino acid 993 (TTG/TGC, haplotype 2S6_B_21) and 1002 (ATT/ATA, haplotype 2S6_B_24) in Clade B; and 1000 + 1016 (respectively GTA/GTG and TCC/TCT in the same haplotype 2S6_A_19) in clade A (Fig 2).

The most common haplotype was the 2S6_B_00 (Clade B), found in 69 out of 79 populations. However, among the seven African populations sampled, only Sedhiou (Senegal) had this haplotype. Also in Clade B, the 2s6_B_04 haplotype was found in Africa (Kenya), Australia and Brazil. Interestingly, the oldest samples (2001–2009) from Brazil contained this haplotype.

In clade A, only six haplotypes were detected. The 2S6_A_03 allele was present in all continents included in the analysis. All haplotypes containing a *kdr* allele were placed in clade A: 2S6_A_01 (1011M), 2S6_A_02 (1016I) and 2S6_A_06 (989P + 1016G), all identical to the 2S6_A_03 haplotype and differing only at the site that defines *kdr*. Haplotype 2S6_A_01 was found in populations from Brazil, Colombia, and Dominica; the 2S6_A_02 in Brazil, Colombia, Haiti, Dominica, Mexico, USA and Senegal; and 2S6_A_06 only in Asian populations: Saudi Arabia, Thailand and Philippines. We did not find any haplotype carrying only the 989P or 1016G alone. The other two haplotypes in clade A, 2S6_A_19 and 2S6_A_22, were exclusive to African localities (Cameroon and Senegal, respectively).

A IIS6 haplotype network analysis shows the three *kdr* haplotypes (2s6_A_01, 2s6_A_02 and 2s6_A_06) directly linked to the 2s6_A_03 haplotype (Fig 2). The 2s6_A_03 haplotype was widely distributed, present in Americas (Tucson, Haiti, Venezuela and Brazil), Africa (Sedhiou and Nairobi), Asia (Cebu) and Pacific (Hawaii), and Australia (S6 Fig). The results of the network analyses together with the wide distribution of the 2s6_A_03 haplotype suggests a single origin, with an haplotype similar to 2s6_A_03 as the likely ancestor of the *kdr* haplotypes with derived SNPs in sites 989, 1011 and 1016.

Similar analyses with sequences of the IIIS6 segment reveled 18 haplotypes between 353 and 354 bp. Nine of the haplotypes only occurred in a single population. Out of these nine, six belonged to African populations (S3 Table). Sequences are available on GenBank (accession numbers: MN602779-MN602796). The greatest variation was observed inside the intron, with 18 variable sites, and 11 SNPs found in the exons (seven synonymous and four non-synonymous (S4 and S5 Figs). The most frequent haplotype (3s6_00) was found in all continents and eleven haplotypes were specific to Africa. Haplotype 3s6_02 and 3s6_03 were found in three (Africa, South America, and Asia) and two (Africa and South America) continents, respectively. Four haplotypes presented non-synonymous substitution: 3s6_01 (1534C, found in Africa, Americas and Asia), 3s6_13 (1520I + 1534C, in Asia), 3s6_11 (1523T, in Haiti) and 3s6_17(1605A, a novel SNP found in Brazil). Contrary to the phylogenetic analyses for the IIS6 segment haplotypes, a ML tree using the IIIS6 segment haplotypes did not provide support for distinct clades (S6 Fig).

The TSC haplotype network analysis for the IIIS6 haplotypes is consistent with a single origin of the 1534C *kdr* mutation, which despite occurring worldwide, it was only found in two IIIS6 haplotypes (3s6_01 and 3s6_13). This network also implies that the Asian 1520I + 1534C haplotype (3s6_13) originated from 3s6_01 (Fig 3). The frequencies of both IIS6 and IIIS6 haplotypes with and without a known *kdr* SNP in each population are represented in Fig 4.

## Reconstructing Na$_v$ evolution by merging information of IIS6 and IIIS6 haplotypes

The variable intron in the haplotypes of the IIS6 segment, which grouped the sequences in clades A and B was usedto determine if the *kdr* mutations in this segment had single or multiple origins. As 1011M, 1016I and 989P + 1016G were present in single haplotypes, there was

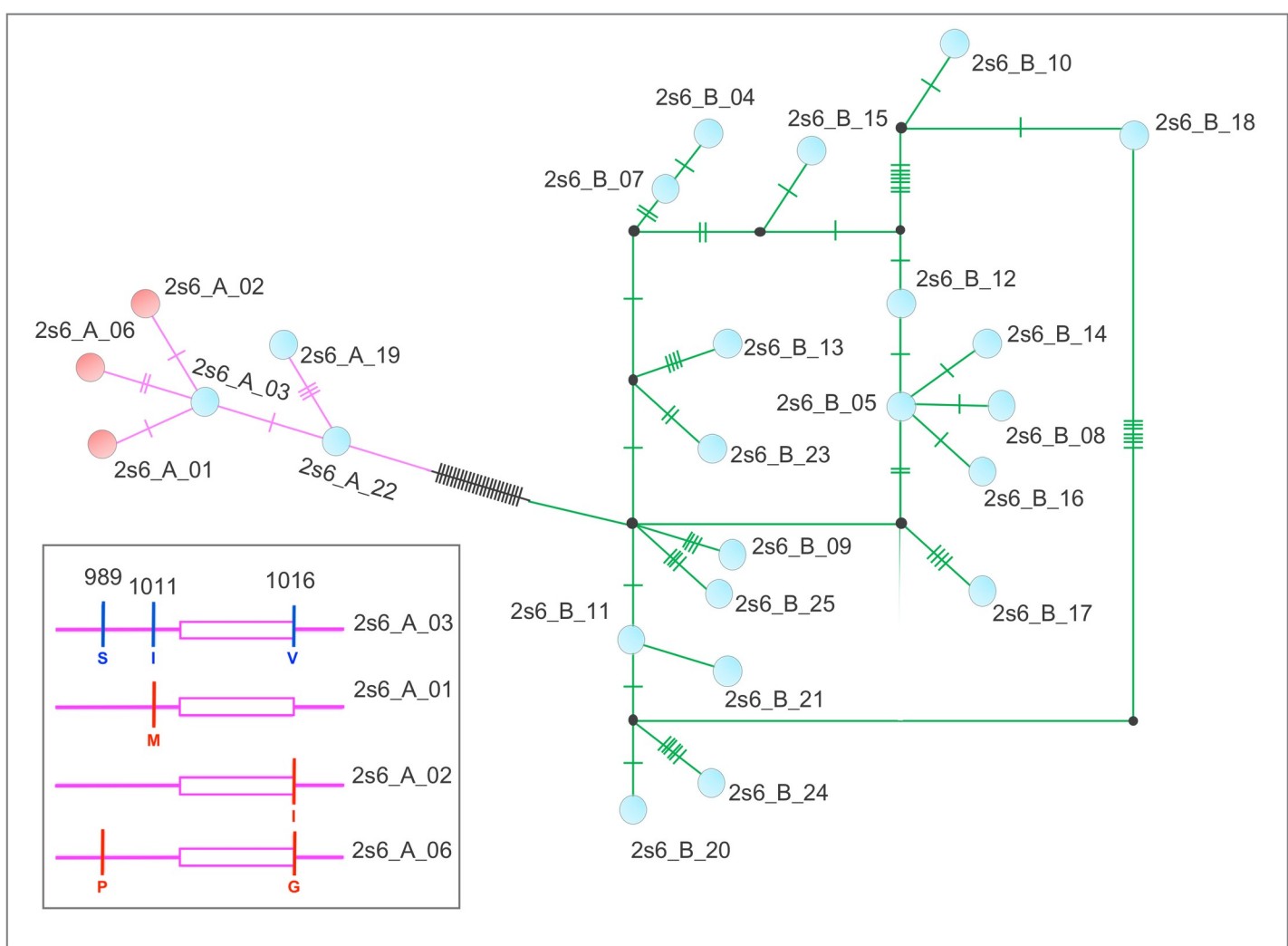

**Fig 2. Haplotype Network of nucleotides corresponding to the IIS6 segment of the voltage gated sodium channel gene of *Aedes aegypti* populations from diverse continents.** The network was designed based on the genealogies calculated by the TCS method [21]. Each dot represents a haplotype, connected by lines in which each tick mark indicates a mutational step between connected haplotypes. Lines are color-coded to indicate Clade A (purple) or Clade B (green). Dot colors indicate haplotypes with (red) or without (blue) *kdr* mutations. A schematic representation of the clade with *kdr* haplotypes is in the box, with *kdr* SNPs in red.

no evidence of multiple origins of these *kdr* mutations. In the IIIS6 segment, the intron was not highly variable and although the *kdr* 1534C was present in two haplotypes, they were identical except for a difference in one nucleotide (3s6_01 and 3s6_13). This is consistent with a single origin for the *kdr* 1534C mutation. We phased IIS6 and IIIS6 to combine the variability from both segments, and thus increase our analysis power. The phasing could only be determined in populations which haplotypes were monomorphic in at least one of the segments, because it is not possible to indicate which haplotypes are on the same chromosome when populations were polymorphic in both segments. Six populations (Puerto Rico, New Orleans, Iguala, Amacuzac and Ribeirão Preto) satisfied this condition of being monomorphic for IIIS6, all presenting the *kdr* 1534C haplotype 3s6_01, and two IIS6 haplotypes: 2s6_B_00 and the *kdr* 1016I 2s6_A_02 (Fig 5A). Therefore, in these cases the only possible phased haplotypes were 2S6_B_00 + 3s6_01 (*Phased*_00B-01) and 2S6_A_02 + 3s6_01 (*Phased*_02A-01). As 2s6_B_00 and 2s6_A_02 are in clades A and B, respectively (see Fig 2), and both phased

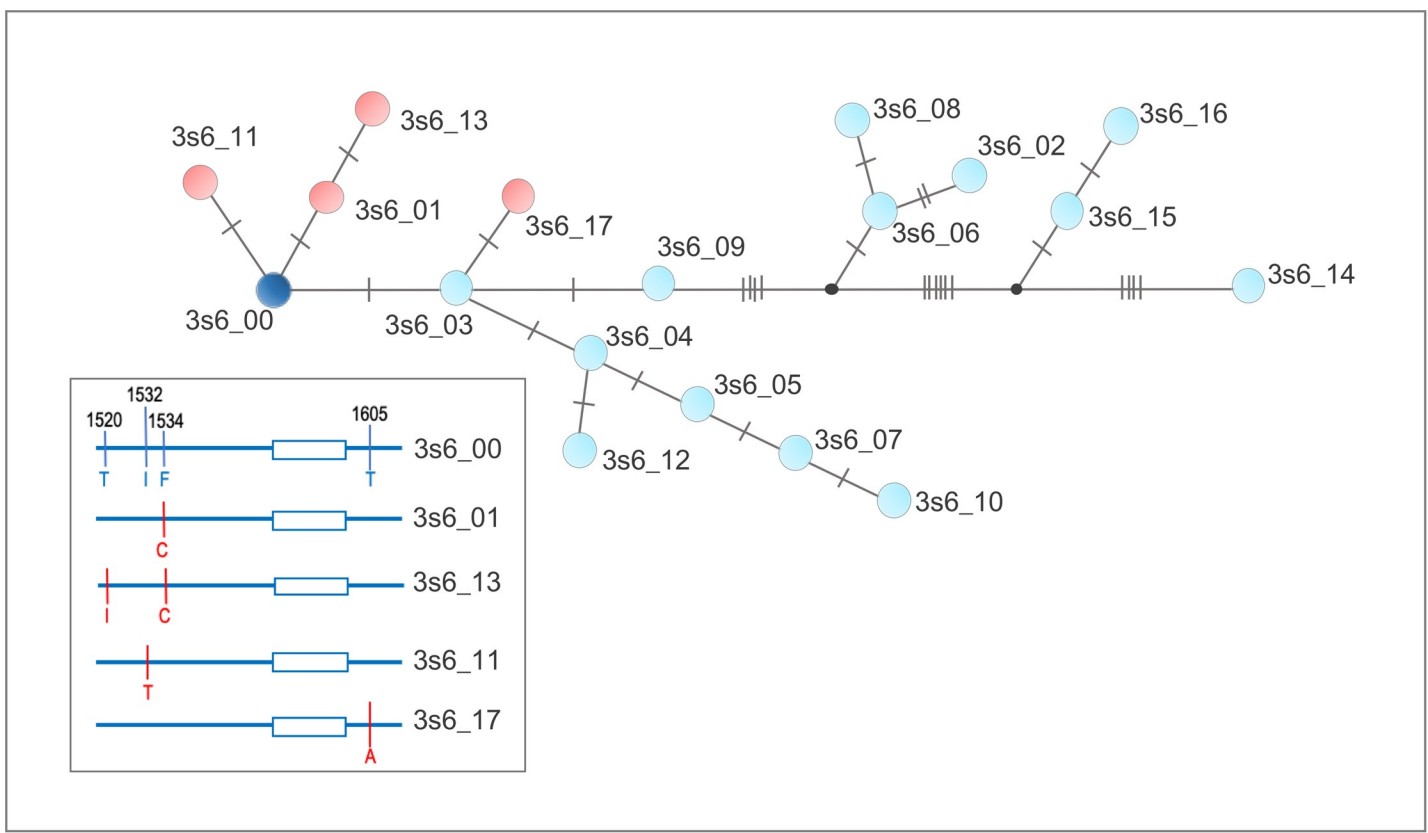

**Fig 3. Haplotype Network of the IIIS6 segment of the voltage gated sodium channel gene of *Aedes aegypti* populations from diverse continents.** The network is the genealogy calculated by the TCS method [21]. Each dot represents a haplotype, connected by lines in which each slash indicates a mutational step between connected haplotypes. Red and blue dots indicate synonymous and non-synonymous substitutions, respectively. The most common haplotype 3s6_00 is in dark blue. A schematic representation with haplotypes containing non-synonymous SNPs is in the box, with SNPs in red.

haplotypes carry the *kdr* 1534C mutation in the IIIS6 segment, evidence suggests that this mutation had at least two independent origins. This information was not available when looking only at the IIIS6 haplotype network alone (Fig 3).

We suggest that the *kdr* 1534C *Phased 00B-01* present in Brazil, Colombia, Mexico, Puerto Rico and USA could have derived from *Phased_00B-00* found in Brazil, USA and Australia (Fig 5B). In addition, we hypothesize that the *Phased_03A-00*, found in USA and Australia, likely originated a *Phased 03A-01*. This should have been the ancestor of the other 1534C haplotypes, which also contained the *kdr* mutations in the IIS6 segment: 1016I (*Phased_02A-01*) in Brazil, Colombia, Mexico, Puerto Rico and USA, and 989P + 1016G (*Phased_06A-01*) in Saudi Arabia, Thailand and Philippines. There was no evidence of multiple origins for the *kdr* mutations in the IIS6 segment (Fig 5C).

In summary, we have inferred that there are at least two *wildtype* (insecticide susceptible) $Na_V$ phased haplotype sources of the *kdr* mutation (insecticide resistant): *Phased_00B-00* and *Phased_*03A-00 (Fig 5), probably originally in Africa.

## Temporal variation

Samples from Brazil from years spanning 2001 to 2015 were used in order to evaluate changes in frequency of the $Na_V$ haplotypes. Five haplotypes in the IIS6 segment were found in these populations: 2s6_B_00, 2s6_B_04, 2s6_A_01, and the *kdr* 2s6_A_02 and 2s6_A_03. We

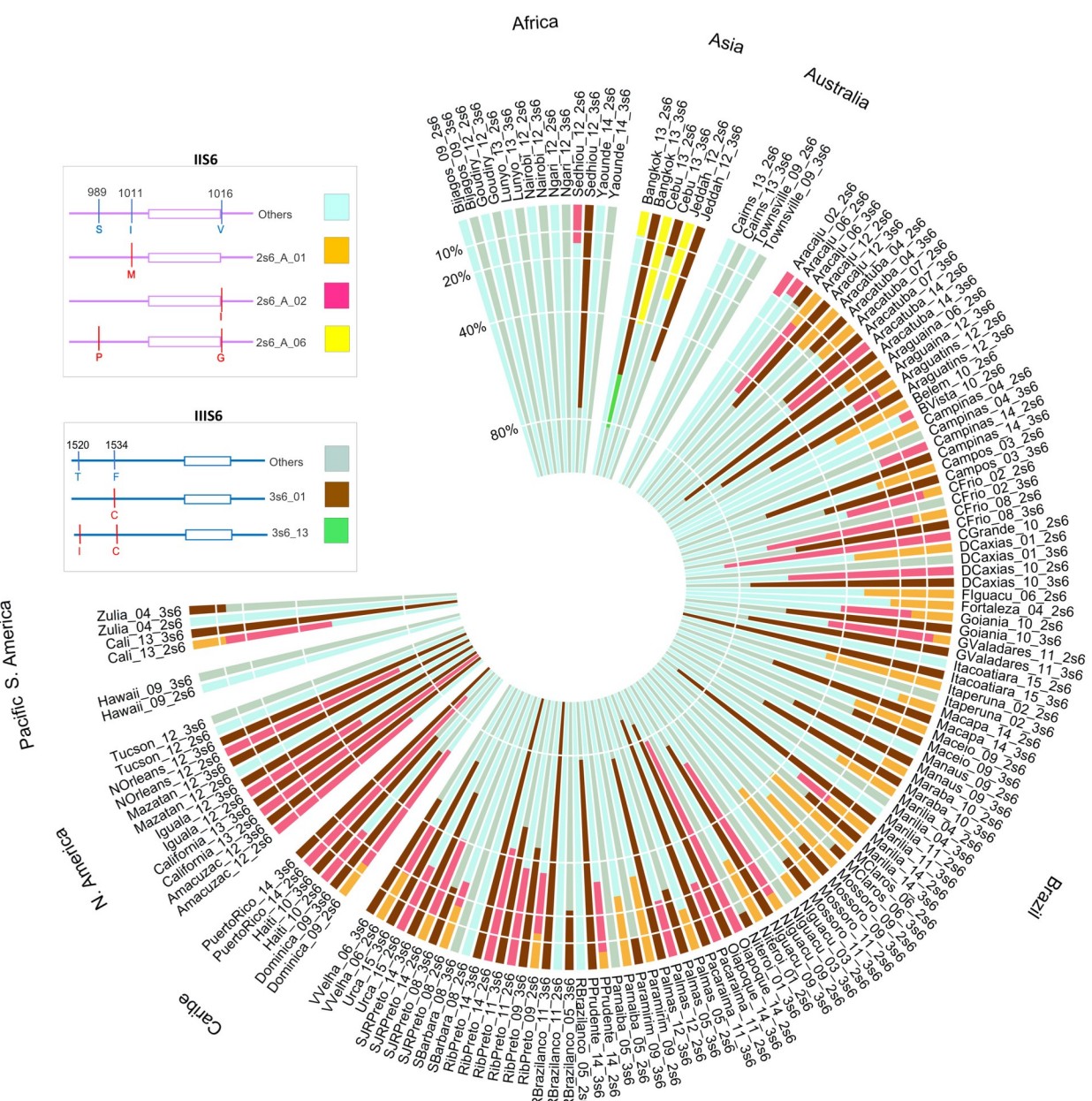

**Fig 4. Frequencies of haplotypes with or without non-synonymous substitutions in IIS6 and IIIS6 Na$_V$ segments of *Aedes aegypti* worldwide populations.** The populations contain one bar for each of the two Na$_V$ segments. In each bar it is presented the frequency of the haplotypes with non-synonymous SNPs and the sum of the remaining haplotypes (wild-type). Populations are grouped according to their continent.

observed that the *kdr* 1016 Ile (2s6_A_02) first appeared in samples from 2002, only achieving higher frequencies from 2007 onwards. On the other hand, the 1011 Met (2s6_A_01) was present during the whole period, however seems to decrease in frequency. Among the non *kdr* haplotypes, 2s6_A_03 and 2s6_A_04 also tended to decrease, whilst 2s6_B_00 was generally the most frequent haplotype (S7 Fig). Five haplotypes were observed in the IIIS6 segment of *Ae. aegypti* populations from Brazil: 3s6_00, 3s6_02, 3s6_03, 3s6_17 and the *kdr* 3s6_01 (1534 Cys). The *kdr* haplotype was present in 2001, at low frequency, with a very clear progressive increase in frequency until 2015, in parallel to the decrease of 3s6_00. The 3s6_02 haplotype

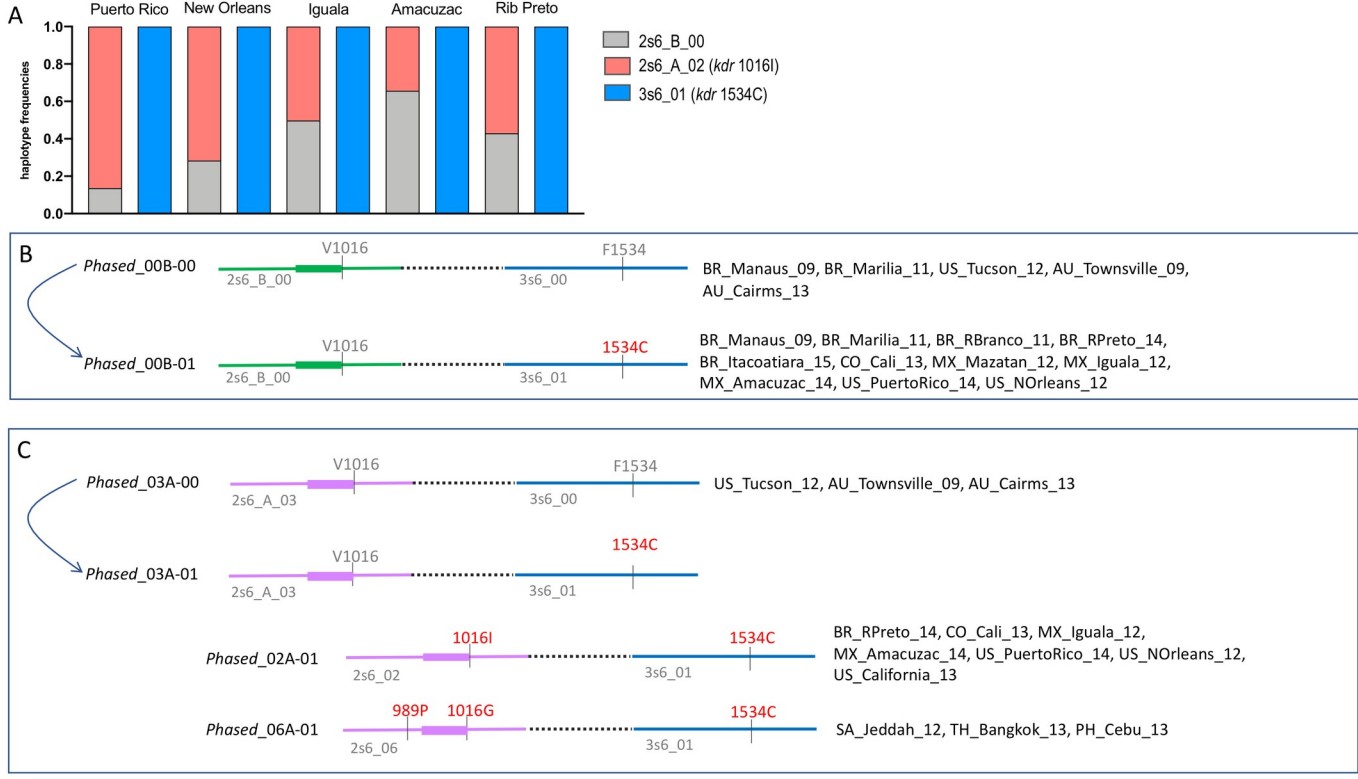

**Fig 5. Evidence for at least two independent origins for the 1534C *kdr* mutation in *Aedes aegypti*.** Bars show the haplotype frequencies of IIS6 and IIIS6 Na$_V$ segments found in five localities that are fixed for 3s6-01 haplotype (**A**). The IIS6 haplotypes from Clades A or B (see Fig 2) are represented as purple and green lines, respectively. Two different origins are proposed for the 1534C *kdr* mutation in **B** and **C** panels. The single origin for the other *kdr* mutations is also suggested in panel **C**. Phased haplotypes are followed by the name of the populations where they were found. *Kdr* SNPs are shown in red.

generally occured at low frequencies, except in a population where was fixed, and the other two haplotypes were rare (S8 Fig). The remarked increase of 3s6_01 makes sense, since it is part of both *Phased*_00B-01 (1534 Cys) and *Phased*_02A-01 (1016 Ile + 1534 Cys), as afore mentioned. This is consistent with the increase of *kdr* resistance in *Ae. aegypti* from Brazil in the last decade and indicates that the *kdr* 1534C from both distinct origins are present in this country.

## Discussion

Molecules that are targeted by neurotoxic insecticides are remarkably conserved among distant taxa, implying the encoded proteins are under selective constraints even in the absence of insecticides [26]. This explains the repeated changes at the classical *kdr* 1014 site (Leu to Phe or Ser) in several insect species, including other mosquitoes in the genera *Culex* and *Anopheles*. In *Aedes*, however, due to a codon constraint, two nucleotide substitutions in the relevant codon are needed to produce the amino acid changes found in other insects, reducing its likelihood [27]. Thus, in *Aedes*, alternative mutations that change the channel conformation making it resistant to pyrethroids without losing its physiological role have arisen. Mutations at Na$_V$ sites 1016 and 1534 produce resistance and are found not only in many populations of *Ae. aegypti*, but also in *Aedes albopictus* [28, 29], a likely result of convergent adaptations to pyrethroids pressure. This indicates that the relevant mutations may arise multiple times in a species and thus the geographic distributions of resistant alleles is not always the result of

migration from a single geographic origin. Our analyses found evidence of multiple origins of the 1534C *kdr* mutation in *Ae. aegypti*, as previously hypothesized [30, 31]. Similar analyses of intronic sequences near the 1014 site also indicated convergent evolution of the *kdr* mutation in several insects, such as *An. gambiae* [15], in the house fly *Musca domestica* [32], the codling moth *Cydia pomonella* [33] and the aphid *Myzus persicae* [34].

## Spatial analyses

Here we found the largest diversity of $Na_V$ haplotypes in Africa with 14 out of 17 and 23 out of 25 of IIS6 and IIIS6 haplotypes, respectively. This higher diversity in African populations was expected since they are older than populations outside Africa [1] and is consistent with the high diversity observed with other genetic markers in Africa [(e.g. microsatellites, [35])]. In addition to being younger, populations outside Africa may be subject to selective sweeps around the $Na_V$ gene due to exposure to insecticides. There is remarkably little data on insecticide resistance in *Ae. aegypti* from Africa [9], in contrast to the information for *Anopheles*. The lack of attention to insecticide resistance in African *Ae. aegypti* may be due to insecticides rarely being employed specifically against this mosquito in Africa. We suggest that exposure could be a byproduct of extensive use against malaria vectors. Among the five African countries from which we had samples, we found a *kdr* haplotype only in Senegal. It is known that Senegal has populations with mixed genetic signature of Aaf and Aaa [35, 36], so it is possible this *kdr* haplotype came back from the Americas.

In Africa, the first detection of *kdr* was in resistant populations from Ghana (West African coast) from samples collected in 2012 and 2014 [37]. The *kdr* 1534C mutation was observed in individuals with scaling patterns usually found in both subspecies (Aaa and AAf). As in our analyses and previous work [38], the IIS6 haplotypes belonged to two distinct clades for IIS6 in those African populations. The *kdr* 1534C African sequences were associated to non-*kdr* IIS6 haplotypes in clade A [37], a result confirmed by our analyses, as the *kdr* 1534C haplotype (in this study haplotype 3s6_01, Fig 3) is also associated with IIS6 haplotypes in Clade A (2s6_A_02: 1016I, Fig 2). However, we also show that this association goes beyond Africa, as it is found in populations from Brazil, Mexico, Puerto Rico, and USA. In addition, the *kdr* 1534C haplotype (3s6_01) was also found with a non-*kdr* IIS6 haplotype in clade B (2s6_B_00) in populations from Brazil, Mexico, Colombia, Puerto Rico and USA. Therefore, the linkage of *kdr* 1534C haplotypes with IIS6 sequences in both distinct clades (Fig 5) indicates that the 1534C *kdr* mutation has emerged independently at least twice. This also explains the high frequency of the haplotype 2s6_B_00 (i.e. without a *kdr* mutation in this fragment), since it is non-randomly associated ("hitchhiked") with the 1534C *kdr* mutation in the IIIS6 segment.

Our samples from Asia were polymorphic for both IIS6 and IIIS6 segments but in these populations we could not infer associations between IIS6 and IIIS6 haplotypes given the impossibility to phase the haplotypes for the two segments. Unlike in the Americas where the 1016I *kdr* mutation generally occurs together with 1534C, 1016G occurs alone or in conjugation with 1534C in Asian populations [39]. Here, we observed only one haplotype with 1016G which also harbored the 989P *kdr* mutation (haplotype 2s6_A_06) found in Saudi Arabia, Thailand, and Philippine populations. Based on the sequence of the intron, this haplotype is in IIS6 clade A, as observed in Taiwanese populations [40]. Additionally, 1534C was linked to IIS6 clade B, i.e. without mutations in the sites 989 and 1016 [40]. This suggests that in Asian populations 1016G and 989P + 1016G haplotypes emerged from a F1534 haplotype. This is different from what was found with American haplotypes, where 1016I emerged from a 1534C haplotype as also suggested by other studies [30, 31]. In *Ae. aegypti* from Saudi Arabia, the 989P + 1016G + F1534 is in high frequency, with a small portion of the triple mutant 989P

+ 1016G + 1534C [41]. This is consistent with 1534C having more than one origin. Likewise, in Sri Lanka although the triple mutant genotype was found, no individual was homozygote for the three mutations [42].

The substitution T1520I with 1534C (haplotype 3s6_13) was found in Thailand samples, and also in Delhi, India [43]. However, it remains to be elucidated whether this haplotype is resistant to pyrethroids or DDT. Electrophysiological assays of distinct *kdr* mutations introduced in an *AaNa$_V$1* cDNA expressed in a *Xenopus* oocytes heterologous system, evidenced that the 1520I alone did not alter the channel sensitivity to pyrethroids and DDT. However, resistance to permethrin was higher with 1520I + 1534C than with 1534C alone [44]. To our knowledge, the non-synonymous substitution I1532T found in the haplotype 3s6_11 in a population from Haiti has not been previously described in *Ae. aegypti*, but interestingly is found in Italian and Chinese populations of *Ae. albopictus* [45].

## Temporal variation

The substitution I1011M (designated here as 2s6_A_01) was found in the oldest collections we studied (1995 and 1998 from Brazil, Guyana, and Martinique) and only in samples from the New World. Neurophysiological assays showed that mosquitoes with that mutation required higher concentrations of pyrethroids to induce repetitive firing compared to susceptible mosquitoes [46]. Further studies confirmed that I1011M was in higher frequency in pyrethroid resistant individuals in a Brazilian population [25]. However, these previous studies examined only the sequence in the IIS6 segment, meaning that it is possible that the resistant allele 1534C in the IIIS6 segment might also have been present. Indeed, the Brazilian samples we examined here from 2001–2005 had the 1534C mutation (haplotype 3s6_01). In addition, 1011M appeared only as an apparent heterozygote in several natural populations from Brazil. This led to the suggestion that it may be a duplication event [47], a possibility that remains to be confirmed. An *Ae. aegypti* line selected to carry this mutation was not resistant to deltamethrin [48]. However, what is certain is that the frequency of 1011M has been decreasing in parallel to an increase of 1016I (1016I + 1534C) over time, as observed in this study (S7 and S8 Figs).

Focusing on the polymorphism at sites 1016 (Val or Ile) and 1534 (Phe or Cys) in *Ae. aegypti* from Mexico collected between 2000 and 2012 [31], in some localities *kdr* mutations increased from almost zero to near fixation in 12 years. The haplotype V1016 + 1534C increased first, but began to decline when the double *kdr* 1016I + 1534C started increasing. In addition, the haplotype 1016I + F1534 was observed at low frequencies in some Mexican populations but not elsewhere. Vera-Maloof et al. (2015) [31] speculated that low fitness of 1016I + F1534 accounts for their rarity. They also suggested that haplotype V1016 + 1534C arose first, followed by 1016I + 1534C with even greater pyrethroid resistance.

A similar scenario was found in Brazil [49]. The relative level of resistance among these haplotypes was confirmed in laboratory studies: a homozygote line for V1016 + 1534C was less resistant to deltamethrin than the double *kdr* 1016I + 1534C in *Ae. aegypti* with homogeneous genetic background [48]. Results observed in our study corroborate the increase of *kdr* frequency in *Ae. aegypti* from Brazil, which in part justify the spread of pyrethroid resistance and support the genotyping tools currently used for surveillance of 1016I and 1534C *kdr* mutations in natural populations [8].

## Conclusions and future directions

Based on our sequencing of two distinct segments of the *Na$_V$* gene in a spatially and temporally diverse group of populations, we show that the 1534C *kdr* mutation circulating in *Ae. aegypti*

populations around the world has emerged at least two distinct times, likely as convergent selection for resistance to DDT and/or pyrethroids. We found no evidence however of multiple origins of other *kdr* mutations: 1016I (Americas and Africa), 989P + 1016G and 1520I + 1534C (Asia). A temporal analysis with *Ae. aegypti* populations from Brazil showed that the *kdr* 1534C from both origins are present, where the V1016 + 1534C haplotype arose first and more recently the double mutant 1016I + 1534C has been expanding in this country. The combination of these results with global genetic population data will help us to better understand how insecticide resistance arises, is selected, and expands in the major worldwide arbovirus vector, *Ae. aegypti*. Future work will aim at further elucidating specific questions such as how insecticide resistance is evolving in complex landscape scenarios, considering genetic structure, migration as well as vector competence in local populations of *Ae. aegypti*.

## Supporting information

**S1 Table. Data about *Ae. aegypti* populations evaluated in this study: localities and year of sampling, and NGS libraries identification.**
(PDF)

**S2 Table. Quantity of filtered sequences obtained for each population.**
(PDF)

**S3 Table. Frequencies of haplotypes from IIS6 and IIIS6 segments in each population.**
(PDF)

**S1 Text. Methods of sequence analyses.**
(PDF)

**S1 Fig. Frequencies of all haplotypes observed in the IIS6 segments of the voltage gated sodium channel gene of *Aedes aegypti* populations from diverse continents.** Haplotypic frequencies are displayed for each population (panel above), indicating continent, country and year of collection. The panel bellow shows the populations from Brazil.
(TIFF)

**S2 Fig. Alignment of haplotypes of the IIS6 segment of the voltage gated sodium channel gene of *Aedes aegypti* populations from diverse continents.** Nucleotides in the exon and intron regions are in upper- and lower-case, respectively. Dots indicate the same nucleotides as in the 2S6_B_00 haplotype, whereas nonsynonymous changes are in red. The amino acid translation is indicated above the alignment, with changes numbered according to the $Na_V$ protein of *Musca domestica*, as usual.
(PDF)

**S3 Fig. Phylogenetic tree of the haplotypes based on nucleotide sequences of the IIS6 segment of the voltage gated sodium channel gene of *Aedes aegypti* populations from diverse continents.** Evolutionary history inferred using Maximum Likelihood with the Tamura-Nei model in MEGA7 [22]. Bootstrap consensus tree (1000 replicates), only. values over 90% are shown. Branch lengths are in scale with the number of substitutions per site. Colored symbols indicate the continent where the haplotypes were found, according to the legend. Haplotypes found in more than one country are in circles and those found exclusively in one country are indicated with a triangular symbol. Haplotypes with non-synonymous substitutions are outlined in red.
(TIF)

**S4 Fig. Alignment of haplotypes of the IIIS6 segment of the voltage gated sodium channel gene of *Aedes aegypti* populations from diverse continents.** Nucleotides in the exon and intron regions are in upper- and lowercase letters, respectively. Dots indicate the same nucleotides as in the 3S6_00 haplotype. Non-synonymous SNPs changes are in red. The amino acid translation is indicated above the alignment, numbered according to the Na$_V$ protein of *Musca domestica*, as reference.
(PDF)

**S5 Fig. Frequencies of all haplotypes observed in the IIIS6 segments of the voltage gated sodium channel gene of *Aedes aegypti* populations from diverse continents.** Haplotypic frequencies are displayed for each population (panel above), indicating continent, country and year of collection. The panel bellow shows the populations from Brazil.
(TIFF)

**S6 Fig. Phylogenetic tree of the haplotypes based on nucleotide sequences of the IIIS6 segment of the voltage gated sodium channel gene of *Aedes aegypti* populations from diverse continents.** This tree is the bootstrap consensus tree (1000 replicates) inferred by Maximum Likelihood and Tamura-Nei model using MEGA7 software [22], rooted on the 3s6_00 haplotype. Bootstraps values over 90% are shown and branch lengths are in scale with the number of substitutions per site. Colored symbols indicate the continent where the haplotypes were found, according to the legend. Haplotypes found in more than one country are in circles and those found exclusively in one country are indicated with a triangular symbol. Red lined squares denote haplotypes with non-synonymous substitutions.
(TIF)

**S7 Fig. Frequencies of Na$_V$ haplotypes found in *Aedes aegypti* populations from Brazil collected spanning 2001 and 2015.** Each dot represents the haplotype frequency for a given population (see S3 Table). The frequencies of the haplotypes from IIS6 segment.
(TIF)

**S8 Fig. Frequencies of Na$_V$ haplotypes found in *Aedes aegypti* populations from Brazil collected spanning 2001 and 2015.** Each dot represents the haplotype frequency for a given population (see S3 Table). The frequencies of the haplotypes from IIIS6 segment.
(TIF)

## Acknowledgments

We would like to thank Carol Mariani (Yale University, USA), for her outstanding technical assistance, José Bento Pereira Lima (IOC/Fiocruz, Brazil) and Maria de Lourdes Macoris (Sucen-Marilia, Brazil) for collaboration in obtaining samples from Brazil, and to Heloiza Diniz (IOC/Fiocruz, Brazil) for the support with the figures in this paper. We are grateful for support from the Coordination for the Improvement of Higher Education Personnel (CAPES).

## Author Contributions

**Conceptualization:** Jeffrey Robert Powell, Ademir Jesus Martins.

**Data curation:** Luciano Veiga Cosme, Ademir Jesus Martins.

**Formal analysis:** Luciano Veiga Cosme, Ademir Jesus Martins.

**Funding acquisition:** Adalgisa Caccone, Jeffrey Robert Powell, Ademir Jesus Martins.

**Investigation:** Luciano Veiga Cosme, Ademir Jesus Martins.

**Methodology:** Luciano Veiga Cosme, Andrea Gloria-Soria, Ademir Jesus Martins.

**Project administration:** Adalgisa Caccone, Jeffrey Robert Powell.

**Resources:** Adalgisa Caccone, Jeffrey Robert Powell, Ademir Jesus Martins.

**Supervision:** Adalgisa Caccone, Jeffrey Robert Powell.

**Validation:** Andrea Gloria-Soria, Adalgisa Caccone, Jeffrey Robert Powell, Ademir Jesus Martins.

**Visualization:** Ademir Jesus Martins.

**Writing – original draft:** Ademir Jesus Martins.

**Writing – review & editing:** Luciano Veiga Cosme, Andrea Gloria-Soria, Adalgisa Caccone, Jeffrey Robert Powell, Ademir Jesus Martins.

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
