## [Decision Letter · Decision Letter 0]

10 Feb 2020

Dear Ademir Jesus Martins,

Thank you very much for submitting your manuscript "Evolution of kdr haplotypes in worldwide populations of Aedes aegypti: independent origins of the F1534C kdr mutation" for consideration at PLOS Neglected Tropical Diseases. As with all papers reviewed by the journal, your manuscript was reviewed by members of the editorial board and by several independent reviewers. The reviewers appreciated the attention to an important topic. Based on the reviews, we are likely to accept this manuscript for publication, providing that you modify the manuscript according to the review recommendations. 

Sincerely,

Patricia Pietrantonio

Guest Editor

Scott Weaver

Deputy Editor

Reviewer's Responses to Questions

**Key Review Criteria Required for Acceptance?**

**Methods**

-Are the objectives of the study clearly articulated with a clear testable hypothesis stated?

-Is the study design appropriate to address the stated objectives?

-Is the population clearly described and appropriate for the hypothesis being tested?

-Is the sample size sufficient to ensure adequate power to address the hypothesis being tested?

-Were correct statistical analysis used to support conclusions?

-Are there concerns about ethical or regulatory requirements being met?

Reviewer #1: The study is hypothesis-driven and the methods are appropriate for testing the hypothesis

Reviewer #2: Methodology including NGS, phylogenical tools, targeted markers and sample size is adequate and well described in the paper (including in supp information)

Reviewer #3: The temporal variation paragraph needs to be clarified. When you mention that sequences from Brazilian populations spanning from 2001 to 2015 were evaluated in terms of frequency variation, it is not clear which frequencies are you referring to. Do you mean assessing if frequencies of different NaV haplotypes increase or decrease over time? If that so, please state it more explicitly.

**Results**

-Does the analysis presented match the analysis plan?

-Are the results clearly and completely presented?

-Are the figures (Tables, Images) of sufficient quality for clarity?

Reviewer #1: The results are well-presented

Reviewer #2: yes the analysis match the analysis plan. Results are clearly presented and well illustrated by graphs to facilite reader understanding. Resolution (quality) of pictures may need to be double check.

Reviewer #3: In the Haplotype analyses section it is not clear what you mean by "Nine haplotypes were specific to a single population, six in Africa." Could you please clarify? Does this mean that the 9 haplotypes were all found within population "6" in Africa? Please clarify the language to make this explicit.

**Conclusions**

-Are the conclusions supported by the data presented?

-Are the limitations of analysis clearly described?

-Do the authors discuss how these data can be helpful to advance our understanding of the topic under study?

-Is public health relevance addressed?

Reviewer #1: The conclusion is supported by their data. 

The limitations of analysis are not discussed `

Reviewer #2: The conclusion supports the data. The discussion section however is a bit too "technical " for me and could be improved to ease reader understanding (mentioning all haplotype names can be boring and difficult to follow). Puting their findings in perspectives of vector control activities in Brazil would be a plus!

Reviewer #3: As part of the conclusions it is stated that future work will aim at further elucidating how insecticide resistance is interfering with genetic structure, migration as well as vector competence in local populations of Ae. aegypti. I would suggest to express this differently because it can be argued that population genetic structure is what actually modulates patterns if insecticide resistance, and vector competence. That is, genetically distinct populations may differ in their potential to develop insecticide resistance as well as in vector competence. Most likely insecticide use and population genetics are likely integrating and affecting vectors' evolution.

**Editorial and Data Presentation Modifications?**

Reviewer #1: (No Response)

Reviewer #2: (No Response)

Reviewer #3: In the Abstract/Methods section replace "populations from 15 countries, including South and North America, Africa, Asia, Pacific, and Australia." with "populations from 15 countries in South and North America, Africa, Asia, Pacific, and Australia.

In the Abstract/Conclusion section replace "These results support that some kdr mutations arise de novo and are regionally distributed in Ae. aegypti. However, our results also suggest for the first time that the 1534C kdr mutation had at least two independent origins." with "These results provide evidence of kdr mutations arising de novo at specific location within the Ae. aegypti geographic distribution. In addition, our results suggest that the 1534C kdr mutation had at least two independent origins. "

In the Author Summary Section: Delete "the" before "residences" in the following sentence: "...the class of insecticide most employed worldwide inside and around the residences" 

In the Author Summary Section: Change "from a DNA collection" to ""from DNA collections" in this sentence: "By evaluating the sequences of two fragments of the NaV gene, obtained from a DNA collection of Ae. aegypti from several countries,..."

In the Author Summary Section: Tell me first that two independent origins were evidenced for the F1534C kdr mutation, found in American, African and Asian populations and then mention the lack of geographic variation in some Kdr mutations (i.e., positive results first).

In the Author Summary Section: Rewrite this sentence: "These results improve the knowledge about insecticide resistance evolution in the mosquito who is presently one of the main actors of global health concerns." to something more like this "Our results clarify insecticide resistance evolution in one of the main vectors of several global diseases"

In the Introduction add "worldwide" to this sentence: "Aedes aegypti is considered one of the most successful invasive species, worldwide.."

In Discussion section: Add scientific name for the codling moth 

In Discussion section: When you say "They found the kdr 1534C mutation in individuals with scaling patterns usually found in both subspecies (Aaa and AAf)." It is no clear what "they" is referring to. Change to passive voice or mention the subject to which "they" is referring to for clarity. Apply this to the whole paragraph.

**Summary and General Comments**

Reviewer #1: Mutations in two transmembrane regions , IIS6 and IIIS6, of the sodium channel protein have previously been shown to be associated with resistance of Aedes aegypti to pyrethroid insecticides. This manuscript reports a haplotype analysis of the genomic regions encoding IIS6 and IIIS6 in Ae. aegypti populations from 15 countries to understand the evolution of these mutations, known as kdr mutations. This study involved a large amount of sequence data as well as haplotype network and phylogenetic analyses. They found 26 and 18 haplotypes for IIS6 and IIIS6, respectively, identified previously identified kdr mutations in some of the haplotypes and confirmed specific geographical distributions of these mutations. Evidence is provided suggesting that one of the mutations, F1534C in IIIS6, arose twice with two different origins. However, origins of the kdr mutations in IIS6 remain unclear.

Data presented seem solid. I have only the following minor comments:

The authors need to point out that I1532T and 1605A have not been shown to be associated with pyrethroid resistance in Ae. aegypti; However, the role of T1520I in pyrethroid resistance has been functionally confirmed in an in vitro expression system (Chen et al., 2019, PloS NTD) 

Page 7, 3rd paragraph: 1523T in “The Asian 1523T +1534C haplotype (3s6_13)” should be “1520I.”

 The resolution of Fig. 4 is poor.

Reviewer #2: Dear Editor,

The authors investigated the origin and dispersion of kdr haplotypes in Ae. aegypti from multiple locations (South and North America, Africa, Asia, Pacific, and Australia) using a combination of NGS and phylogeny approaches. The authors showed that all mutations seemed to be “regionally” distributed while the 1534C kdr mutation seems to be widely distributed worldwide. Their results suggest that most of kdr mutations had single origin (ie “de novo” mutation) and then spread regionally while the 1534C kdr mutation had at least two independent emergence. Although the paper is not always “easy” to follow (especially the discussion part that contains a lot of haplotype names), this study provides new insight into the selection and spread of insecticide resistant markers in Aedes mosquito and has practical implications for designing more efficient Insecticide Resistance containment plan. The paper is suitable for publication in Plos NTD after minor revision.

Minor revisions;

The discussion is too “technical” because it contains many (unecessary?) abbreviations and haplotype names! The authors should make efforts to summarize the information’s instead of repeating the results. I would advise them also to discuss the results in perspective of vector control activities (eg history of insecticide use in Brazil and occurrence and spread of kdr mutations, how to use their data to better implement IRM ?, etc) 

P10, Third paragraph. Interestingly the I1011M mutation seems to decrease over time in Brazil. Since pyrethroid resistance is increasing in Aedes aegypti, do the authors suggest that the I1011M has no link with PYR resistance? If the answer is yes, the authors should be more explicit on this matter. 

P20, fourth paragraph. The absence of double homozygotes for kdr mutations at position 1016 and 1534 has been previously observed in South East Asia especially in Thailand, Lao and Myanmar (Marcombe et al 2019; Kawada et al 2010). The negative genetic association between the 1016 and 1534 kdr indicate that the two genes are not independent from each other. How did the authors have addressed this issue in the analysis? 

Knowing that the authors have mosquito samples from multiple origin (regions) and time scale, would it be possible to determine where and when the 1534C mutation firstly emerged (eg Africa, Latin America, Asia?)

I wonder why the authors didn’t combine NGS plus classical population genetics tools to determine the possible “routes” of resistance spread ? Please justify.

Reviewer #3: My only suggestion to improve the presentation of this research is to include a table that shows a summary of the number of individuals use per population or at least per country. That information can be currently obtained by adding samples in the tables provided. A summary table will be useful to gather this information at a glance

PLOS authors have the option to publish the peer review history of their article (what does this mean?). If published, this will include your full peer review and any attached files.

Reviewer #1: No

Reviewer #2: No

Reviewer #3: No
---

## [Editor Report · Decision Letter 1]

13 Mar 2020

Dear Dr Martins,

We are pleased to inform you that your manuscript 'Evolution of kdr haplotypes in worldwide populations of Aedes aegypti: independent origins of the F1534C kdr mutation' has been provisionally accepted for publication in PLOS Neglected Tropical Diseases.

Best regards,

Scott C. Weaver

Deputy Editor

Scott Weaver

Deputy Editor

---

## [Editor Report · Acceptance letter]

7 Apr 2020

Dear Dr Martins,

We are delighted to inform you that your manuscript, "Evolution of kdr haplotypes in worldwide populations of Aedes aegypti: independent origins of the F1534C kdr mutation," has been formally accepted for publication in PLOS Neglected Tropical Diseases.

Best regards,

Serap Aksoy

Editor-in-Chief

Shaden Kamhawi

Editor-in-Chief
